# Preparation and Antioxidant Activity of Chitosan Dimers with Different Sequences

**DOI:** 10.3390/md19070366

**Published:** 2021-06-25

**Authors:** Wentong Hao, Kecheng Li, Yuzhen Ma, Rongfeng Li, Ronge Xing, Huahua Yu, Pengcheng Li

**Affiliations:** 1CAS and Shandong Province Key Laboratory of Experimental Marine Biology, Center for Ocean Mega-Science, Institute of Oceanology, Chinese Academy of Sciences, Qingdao 266071, China; haowentong0723@163.com (W.H.); mayuzhen2018@163.com (Y.M.); rongfengli@qdio.ac.cn (R.L.); yuhuahua@qdio.ac.cn (H.Y.); pcli@qdio.ac.cn (P.L.); 2Laboratory for Marine Drugs and Bioproducts, Pilot National Laboratory for Marine Science and Technology (Qingdao), No. 1 Wenhai Road, Qingdao 266237, China; 3University of Chinese Academy of Sciences, Beijing 100049, China

**Keywords:** chitosan dimers, sequence, separation, biological method, antioxidant activity

## Abstract

As a popular marine saccharide, chitooligosaccharides (COS) has been proven to have good antioxidant activity. Its antioxidant effect is closely related to its degree of polymerization, degree of acetylation and sequence. However, the specific structure–activity relationship remains unclear. In this study, three chitosan dimers with different sequences were obtained by the separation and enzymatic method, and the antioxidant activity of all four chitosan dimers were studied. The effect of COS sequence on its antioxidant activity was revealed for the first time. The amino group at the reducing end plays a vital role in scavenging superoxide radicals and in the reducing power of the chitosan dimer. At the same time, we found that the fully deacetylated chitosan dimer **DD** showed the strongest DPPH scavenging activity. When the amino groups of the chitosan dimer were acetylated, it showed better activity in scavenging hydroxyl radicals. Research on COS sequences opens up a new path for the study of COS, and is more conducive to the investigation of its mechanism.

## 1. Introduction

Reactive oxygen species is a one-electron reduction product of a type of oxygen in the body. A high concentration of reactive oxygen species can damage proteins, lipids and DNA, which leads to oxidative stress, an imbalanced state in the body [1,2,3]. Oxidative stress is a major reason for the initiation and progression of many diseases, including cardiovascular diseases [4], Parkinson’s disease [5], atherogenesis [6], neurodegeneration [7] and cancer [8]. Antioxidants can scavenge free radicals or retard the progress of many chronic diseases as well as lipid peroxidation. The most common antioxidants are phenolic compounds, such as butylated hydroxyanisole (BHA), butylated hydroxytoluene (BHT), tert-butylhydroquinone (TBHQ) and propyl gallate (PG). However, these antioxidants are toxic to a certain extent, and nowadays people are more inclined to look for non-toxic, efficient, natural antioxidants [9,10].

Biological polysaccharides have attracted widespread attention in recent years for their non-toxicity, availability and good antioxidant activity. For example, the polysaccharide from ginseng has relatively high antioxidant activity in 2,2-diphenyl-1-picrylhydrazyl (DPPH) radical scavenging. Polysaccharides derived from marine organisms also exhibit strong antioxidant activity, such as polysaccharides from marine algae [11,12,13], polysaccharides from sea cucumber [14] and chitosan from shrimp and crab shells. Chitosan is a linear polysaccharide made of β-1,4 linked d-glucosamine (GlcN, **D**) and β-1,4-linked *N*-acetyl-d-glucosamine (GlcNAc, **A**) (Capital letter A in bold stands for the *N*-acetyl-d-glucosamine unit and capital letter D in bold stands for the d-glucosamine unit; the glycosidic bond between the monosaccharide units was omitted in this study), which is good in scavenging hydroxyl radicals and has chelating abilities on ferrous ions [15]. Chitooligosaccharides (COS) are the oligomers of chitosan and are a non-toxic, efficient, novel antioxidant as well. Chitosan and COS are described by their molecular weight (MW) or degree of polymerization (DP), degree of acetylation (DA), pattern of acetylation (PA) or sequence [16,17]. COS have a smaller MW and better water solubility than chitosan, which made its antioxidant activity having been extensively studied. Previous studies showed COS can effectively scavenge DPPH free radicals and protect human embryonic liver cells from oxidative stress induced by H_2_O_2_ [18]. COS can also effectively protect human nerve cells from oxidative stress induced by copper ions [19]. Experiments in vivo also proved COS can significantly increase the total antioxidant capacity and superoxide dismutase (SOD) activity of rats, and significantly reduce the content of malondialdehyde (MDA) in the serum of inflammatory rats [20].

In addition, the antioxidant activity of COS is dependent on its chemical structure, such as the MW or DP and DA [21,22]. Marian et al. found COS with a DP ranging from 3 to 7 have better antioxidant activity than COS with a DP of 3–17 and 3–24 [21]. Our group further studied the antioxidant activity of each COS with a single DP and found COS with a low DP showed a better effect of scavenging hydroxyl radicals and reducing power than COS with a high DP [23]. Experiments in vivo also proved that COS with a low DP has better antioxidant activity. COS (DP = 1–5) can elevate levels of high-density lipoprotein cholesterol, the activity of lipoprotein lipase, hepatic lipase, superoxide dismutase and glutathione peroxidase [24]. Meanwhile, the antioxidant activity of COS is also changing with the DA. Jae-Young Je et al. found COS with a relatively lower DA showed a higher radical scavenging activity on DPPH, hydroxyl radicals, carbon-centered radicals and superoxide radicals [25,26]. However, Li et al. separated three chitotrioses with a different DA and got inconsistent results. They found that chitotriose with a higher DA exhibited stronger antioxidant activity [27].

It is worth noting that COS with a single DP and the same DA may still contain various isomers of different sequences and it is difficult to separate these isomers. Therefore, the relationship between the structure and antioxidant activity of COS still remains ambiguous to some extent. Recently, more studies have shown that the sequence of COS has an important influence on its biological activity [28,29]. However, the effect of sequence on the antioxidant activity of COS is currently very unclear. Considering COS with a low DP has a better effect on antioxidant activity, the antioxidant activity of four chitosan dimers with different sequences was studied in this paper. Firstly, three different sequences of chitosan dimers (**AA**/**AD**/**DA**) were prepared by size exclusion chromatography (SEC) and enzymatic deacetylation. In addition, **DD** was prepared previously by ion exchange chromatography. Then, the in vitro antioxidant activities of these four dimers were performed to reveal their structure–activity relationship. These results would have important significance for the development of COS antioxidants. 

## 2. Results

### 2.1. Preparation and Characterization of the Chitin Oligomers

Chitosan dimers have four possible sequences, namely, **AA**, **AD**, **DA** and **DD**. It is difficult to separate these four different sequences from a natural COS mixture. Their chemical synthesis is also time-consuming and troubling due to multiple protection and deprotection processes. In this study, we prepared these chitosan dimer isomers by enzymatic deacetylation. Firstly, the *N*-acetyl chitooligomers mixture ((GlcNAc)_1–6_) was separated by SEC to obtain the **AA** that can be the substrate of subsequent enzymatic hydrolysis. As is shown in Figure 1, we collected six fractions by measuring the absorbance at 210 nm of the elution. According to the principle of SEC, these six fractions, F1–F6, are expected to correspond to a chitin hexamer (**A**6), chitin pentamer (**A**5), chitin tetramer (**A**4), chitin trimer (**A**3), chitin dimer (**AA**) and *N*-acetylglucosamine. Subsequently, F5 was selected to be further analyzed through Electron Spray Ionization/Mass Spectrometry (ESI-MS) (Figure 2A) and High-Performance Liquid Chromatography (HPLC) (The result is shown in Section 2.3). The main peak of F5 is at 425.18, which exactly corresponds to the *m*/*z* value of the [M + H] ^+^ ion-peaks of **AA**, suggesting that fraction F5 is indeed a chitin dimer (**AA**), and its purity can reach 98%. The obtained **AA** can be used as the substrate of the enzymatic hydrolysis to prepare the other two chitosan dimers with complex sequences. 

### 2.2. Preparation and Characterization of Two Heterogenous Sequences of the Chitosan Dimer 

With **AA** as the substrate, we used two chitin deacetylases to prepare the other two chitosan dimers. The chitin deacetylase NodB only deacetylates the non-reducing *N*-acetyl glucosamine residue of COS [30]. At the same time, VcCOD only acts on the second acetyl group at the non-reducing end [31]. Then we heterologously expressed these two enzymes in *E. coli* and used them to deacetylate **AA**, producing two heterogenous chitosan dimers N and V. The products of the enzymatic hydrolysis were analyzed by ESI-MS. As shown in Figure 2, the molecular masses of the N-chitosan dimer and V-chitosan dimer are the same. The main peak of 383 corresponds to the [M + H] ^+^ ion peaks of **AA** by loss of 42 Da (exactly the molecular weight of an acetyl group). Therefore, we can infer that **AA** loses an acetyl group after the enzymatic hydrolysis to produce **DA** or **AD**, but the sequences need further characterization.

Considering the overlapping *m*/*z* values for the ions of identical monosaccharide compositions, we cannot distinguish **AD** or **DA** only by the ESI-MS spectrum. Thus, we introduced a tag at the reducing end of the N-chitosan dimer and V-chitosan dimer, as shown in Figure 3A [27]. In the case of 2-aminoacridone (amac) derivatives, the mass increment of 194 Da allows for clear identification of the Y-type ions. Figure 3B,C depicts the MS/MS spectrum of the [M + H] ^+^ ion of *m*/*z* 577.16 of the derivatized V-chitosan dimer and derivatized N-chitosan dimer. The main peak of the derivatized V-chitosan dimer is 374.17, corresponding to the *m*/*z* value of the **D**-amac fragment (Figure 3B). So, it is certain that the sequence of the V-chitosan dimer is **AD**. In the same way, the Y-type fragment ion is observed at *m*/*z* 416.18 in Figure 3C, which corresponds to **A**-amac. Therefore, it is also certain that the sequence of the N-chitosan dimer is **DA**. In this way, we got the other two different sequences of the chitosan dimer.

### 2.3. High-Performance Liquid Chromatography (HPLC) Analysis of Chitosan Dimers

Since we have all the sequences of the chitosan dimers (**DD** was separated by ion-exchange chromatography, as reported previously in our laboratory [32]), we further used the hydrophilic interaction liquid chromatogram to determine their purity. Their characters are summarized in Table 1 and their HPLC spectrograms are listed in Figure 4. As the acetyl group is removed, the affinity between the chitosan dimer and the column is stronger, which leads to a delay in its retention time. At the same time, we found that the retention time of **DA** and **AD** are not the same, verifying that they are two different sequences again. The purity of all four chitosan dimers can reach 98% or even higher.

### 2.4. Antioxidant Activity of the Chitosan Dimers

COS has been demonstrated to exhibit free radical scavenging activities in previous reports and the activity of COS depends on its MW and DA [22,25]. In this study, we investigated the antioxidant activity of four chitosan dimers to explore the role of sequence on antioxidant activity, including hydroxyl radicals, superoxide radicals, DPPH scavenging activity and reducing power. The result is as follows:

Superoxide radicals produced by metabolic processes are considered to be “primary” ROS, which can further interact with other molecules to produce “secondary” ROS [1]. It can attack biological macromolecules, such as lipids, proteins, nucleic acids and polyunsaturated fatty acids, to make them cross-chain or break, causing damage to cells. The superoxide radical scavenging activities of the four chitosan dimers are shown in Figure 5A. The scavenging effects of the four chitosan dimers are all concentration-dependent. **AD** has the best activity compared to the others, whose scavenging effect exceeds 50% at 0.5 mg/mL and reached 75.6% when its concentration was 1.5 mg/mL. At the same time, the scavenging effects of **DA** and **AA** are low, even at a high concentration. Even if the concentration exceeds 1.5 mg/mL, the scavenging effect of these two chitosan dimers is still lower than 15%. The scavenging effect of the four chitosan dimers was in the order of **AD** > **DD** >> **DA**, **AA**, which suggested the chitosan dimer has a better scavenging effect of the superoxide radical when its reducing end is the amino group. 

The amino group in the reducing end is also crucial in the reducing power of the chitosan dimers. As shown in Figure 5B, the high absorbance at 700 nm means a strong reducing power. The reducing power of **DD** and **AD** are increased with concentration while the absorbance of **DA** and **AA** are almost unchanged. However, what is different from the ability to scavenge superoxide anions is the reducing power of **DD** being stronger than **AD**. The reducing power of the chitosan dimers is in the order of **DD** > **AD** > > **DA**, **AA**, which suggests the amino group at the reducing end also plays a decisive role in the reducing power of the chitosan dimer. 

The hydroxyl radical is the neutral form of the hydroxide ion. The hydroxyl radical has a high reactivity, making it a very dangerous radical [1]. The hydroxyl radical scavenging activities of the four chitosan dimers are shown in Figure 5C. The scavenging effect of all the four chitosan dimers are also concentration-dependent. The scavenging effect of the four chitosan dimers was in the order of **AA** > **DA** >> **AD**, **DD**. It seems that the acetylation of the amino group is important for scavenging hydroxyl radicals. The effect of **DA** is obviously better than **AD** at a high concentration, suggesting that the acetyl group at the reducing end may make the activity better. 

DPPH is also a common index to evaluate antioxidation in vitro as a stable organic free radical. However, we found the chitosan dimers cannot scavenge DPPH at low concentrations, except **DD**. The scavenging effects of **AA**, **AD** and **DA** remained relatively low although their concentration has reached 4 mg/mL and a differences cannot be seen between the low and high concentration. The scavenging effect of **DD** are concentration-dependent within our setting concentrations and showed a good DPPH scavenging effect. The scavenging effect of **DD** can reach 64% when its concentration is 4 mg/mL.

## 3. Discussion

In this study, we separated the *N*-acetyl chitooligomers mixture firstly to obtain **AA** as substrate and then **AA** was deacetylated with the specificity of deacetylase (NodB, VcCOD) to obtain two pure chitosan dimers with different sequences. Consequently, four chitosan dimers were prepared, and their structures were characterized. Then we investigated their antioxidant activity and expected to reveal the influence of the sequence on the antioxidant activity and clarify the structure–activity relationship. Our results suggest that the amino groups in the chitosan dimers is important for scavenging several free radicals. For example, the chitosan dimer **DD** exhibit a high scavenging effect of DPPH. The activities of the other three chitosan dimers are not very good, but **DA** and **AD** are still better than **AA**. It means the scavenging effect of the chitosan dimers on DPPH is mainly based on their amino groups. When there are no free amino groups in the disaccharide, there is almost no activity of scavenging DPPH, even at a high concentration. As for **DA** and **AD**, it might be difficult to access the amino group due to the large steric hindrance of DPPH. Previous studies have also shown that the free amino groups in COS plays an important role in scavenging free radical, which is in line with our results [25,33]. Apart from this, we first prove that not only the presence of amino groups is very important for scavenging free radicals, but the position of the amino groups (which is the sequence) also plays a vital role. For example, chitosan dimers with the amino group at the reducing end (**DD** and **AD**) is favorable to scavenge superoxide radicals and enhance the reducing power of the chitosan dimers. On the other hand, the ability of a chitosan dimer to scavenge hydroxyl radicals displayed a different trend. For the hydroxyl radical, the effect of **DA** is obviously better than **AD** at a high concentration, suggesting that the acetyl group at the reducing end makes the activity better. Additionally, the scavenging hydroxyl radical effect of the four chitosan dimers was in the order of **AA** > **DA** >> **AD**, **DD in high concentration**. It seems that the acetylation of the amino group is important for scavenging hydroxyl radicals, which is consistent with previous reports on the antioxidant activity of COS with different degrees of acetylation [19]. Meanwhile, the difference between the four chitosan dimers in scavenging hydroxyl radicals is actually the smallest among these indicators, which also shows that the sequence has the smallest effect on scavenging hydroxyl radicals.

In summary, the antioxidant activity of COS depends not only on its DP and DA, but also on its sequence, as shown in this study. The effect of the chitosan dimers to scavenge different free radicals is different. This maybe related with its mechanism and we are not yet fully confident to draw more specific conclusions. The mechanism of how it works is expected to be carried out further. 

## 4. Materials and Methods 

### 4.1. Material

*N*-acetyl chitooligomers ((GlcNAc)_1–6_) were purchased from the Tokyo chemical industry Co., Ltd (Tokyo, Japan). The NodB sequence (NCBI acc. No. AJW76244.1) and VcCOD sequence (NCBI acc. No. AAF94439.1) were sourced from the NCBI database and provided to Sangon Biotech (Shanghai, China) Co., Ltd. for gene optimization and synthesis. We fused the optimized gene to a HIS6-tag coding sequence for subsequent purification. *E. coli* competent cells were purchased from Sangon Biotech (Shanghai, China) Co., Ltd. and used to express chitin deacetylase. Nitrotetrazolium blue chloride (NBT), phenazine methosulfate (PMS) and nicotinamide adenine dinucleotide-reduced (NADH) were purchased from Sigma Chemicals Co. All other chemicals and reagents were analytical grade.

### 4.2. Separation of N-Acetyl Chitooligomers by Bio Rad P10 

*N*-acetyl chitooligomers (100 mg) was dissolved in 1 mL deionized water and then filtered with a microporous membrane (0.45 μm) to obtain a clear solution. The filtrate was loaded onto a Bio Rad P10 (100 cm × 2.6 cm) column using deionized water as the mobile phase at a flow rate of 0.2 mL/min. Fractions were collected and monitored by a Nanodrop 2000 at 210 nm.

### 4.3. Biological Preparation of the Four Chitosan Dimers 

*E. coli* strain BL21 (DE3) harboring pET-22b (+) NodB plasmid or pET-22b (+) VcCOD plasmid was cultured in LB broth. Isopropyl β-D-thiogalactopyranoside (IPTG) was added at a final concentration of 0.1 mM to induce recombinant gene expression. Cells were lysed with a Cell Disruption System. The protein suspension was submitted to Immobilized Metal ion Affinity Chromatography (IMAC) using a Ni^2+^-nitrilotriacetate-agarose resin. All elution fractions were analyzed by SDS-PAGE. The protein concentration was determined by the Bradford method with BSA as a standard [34,35]. **AA** and **DD** can be obtained by SEC and ion-exchange separation [32]. **DA** and **AD** can be obtained by NodB and VcCOD, separately, using **AA** as the substrate. The reactions of **DA** and **AD** were carried out at 37 °C and 180 rpm overnight with an enzyme:substrate ratio of 1:20. **AA** was incubated with NodB in a 20 mM MOPS buffer [34] and with VcCOD in a 10 mM ammonium carbonate buffer [36], separately. The chitosan dimer products were then loaded onto a Sephadex G10 (100 cm × 2.6 cm) column to get rid of the salt and protein. Fractions were collected and monitored by HPLC.

### 4.4. Reductive Amination of Chitooligosaccharides with 2-Aminoacridone (Amac)

This reaction was performed essentially as described by Bahrke et al. [37]. In total, 1 mg chitosan dimer was dissolved in 20 μL 0.1 mol/L solution of amac in acetic acid/DMSO (3:17, *v/v*), and then 20 μL sodium cyanoborohydride (1 mol/L) was added. Subsequently, the mixture was heated in the dark at 90 °C for 30 min. The resulting solution was lyophilized for ESI MS/MS.

### 4.5. HPLC and ESI-MS Analysis

The chitosan dimers were analyzed by hydrophilic interaction liquid chromatography using an LC-2030C 3D Plus HPLC system (SHIMADZU, Kyoto, Japan) with an evaporative light scattering detector (Essentia ELSD-16). Chromatography was performed on a Click Maltose column (4.6 mm × 150 mm, 5 μm), using binary mobile phases (acetonitrile and ammonium formate buffer) stepwise at a flow rate of 1.0 mL/min and with the column temperature at 30 °C.

The chitosan dimers were analyzed by LTQ Orbitrap XL (Thermo Fisher Scientific, Shanghai, China. The ESI source voltage was 4 kV; sheath gas was 10 arbs. unit; auxiliary gas was 0 arb. unit; sweep gas was 0 arb. unit; and capillary temperature was 275 °C. Samples were analyzed in the FTMS scan mode at a resolving power of 30,000 at *m*/*z* 400. An isolation width of 4 amu was used and precursors were fragmented by CID with a normalized collision energy of 35 V and an activation time of 10 ms. The maximum injection time was set to 100 ms with two micro scans for the MS mode and to 1000 ms with one micro scan for the MS/MS mode. The mass range in FTMS mode was from *m*/*z* 100 to 1500. The data analyses were performed using XCalibur software v2.2 (Thermo Fisher Scientific, (version2.2, Shanghai, China)). An external calibration for mass accuracy was carried out before the analysis.

### 4.6. Superoxide Radical Scavenging Assay

The superoxide radical scavenging ability of the chitosan dimers was modified by the method of Nishikimi, Appaji Rao and Yagi [38]. The reaction mixture, containing 1.5 mL chitosan dimer sample, 0.5 mL PMS (30 μM), 0.5 mL NADH (338 μM) and 0.5 mL NBT (72 μM) in a Tris–HCl buffer (16 mM, pH 8.0), was incubated at room temperature for 5 min in the dark and the absorbance was measured at 560 nm. A 1.5 mL Tris–HCl buffer instead of the sample was the control group; a 1.5 mL Tris–HCl buffer instead of the sample and 0.5 mL Tris–HCl buffer instead of NADH was the blank group. The capability of scavenging superoxide radicals was calculated using the following equation:(1)$$\mathrm{Scavenging}\;\mathrm{effect}\;\left(\%\right)=\frac{A_{\mathrm{control}}-{\;A}_{\mathrm{sample}}}{A_{\mathrm{control}}-{\;A}_{\mathrm{blank}}}\times 100$$

### 4.7. Measurement of Reducing Power

The reducing power was determined by the method of Yen and Chen [39]. Briefly, 2 mL chitosan dimer sample in phosphate buffer (0.2 M, pH 6.6) was incubated with 1 mL potassium ferricyanide (1%, w/v) at 50 °C for 20 min. The reaction was terminated by 1 mL trichloroacetic acid solution (10%, w/v). Then the solution was mixed with 1.5 mL ferric chloride (0.1%, w/v) and the absorbance was measured at 700 nm. High absorbance means a strong reducing power.

### 4.8. Hydroxyl Radical Scavenging Assay

Hydroxyl radicals are produced based on the Fenton reaction. Hydroxyl radicals can bleach safranine O. Antioxidants will provide protons or electrons to the hydroxyl radicals to stabilize the free radicals and reduce the combination of hydroxyl radicals and safranine O, so that the system has a certain absorbance increase at 520 nm. The hydroxyl radical scavenging assay was modified by the method of Guo et al. [40]. A total of 1 mL chitosan dimer sample was incubated with 0.5 mL EDTA–Fe^2+^, 1 mL safranine O and 1 mL H_2_O_2_ (3%) in potassium phosphate buffer (150 mM, pH 7.4) for 30 min at 37 °C The absorbance of the mixture was measured at 520 nm. A total of 1 mL distilled water instead of the sample was the blank group; 1 mL distilled water instead of the sample and 1 mL potassium phosphate buffer instead of H_2_O_2_ was the control group. The capability of scavenging hydroxyl radicals was calculated using the follow equation:(2)$$\mathrm{Scavenging}\;\mathrm{effect}\;\left(\%\right)=\frac{A_{\mathrm{sample}}-{\;A}_{\mathrm{blank}}}{A_{\mathrm{control}}-{\;A}_{\mathrm{blank}}}\times 100$$

### 4.9. DPPH Scavenging Assay

DPPH is a relatively stable free radical, which presents as dark purple in ethanol [41]. When there is an antioxidant, the color will fade. The chitosan dimer sample is mixed with 60 μmol DPPH ethanol solution, the system reacts for 20 min in the dark and the absorbance is measured at 517 nm. A total of 1 mL distilled water instead of the sample was the blank group; 2 mL ethanol instead of the DPPH was the control group. The capability of scavenging DPPH radicals was calculated using the following equation:(3)$$\mathrm{Scavenging}\;\mathrm{effect}\left(\%\right)=\left(1\;-\frac{A_{\mathrm{sample}}-{\;A}_{\mathrm{control}}}{A_{\mathrm{blank}}}\right)\times 100$$

### 4.10. Statistical Analysis

The data are presented as the mean ± SD, followed by Duncan’s multiple-range tests. Differences were considered to be statistically significant if *p* < 0.05.

## 5. Conclusions

In this study, we separated the chitin dimer (**AA**) and further prepared two other chitosan dimers (**DA** and **AD**) by enzymatic hydrolysis using **AA** as the substrate. The antioxidant activities of four different sequences of the chitosan dimer were studied. In the end, we found that the antioxidant activity of COS depends not only on its DP and DA but also on its sequence. In general, **DD** has a better ability to scavenge superoxide radicals and DPPH, and also exhibit a better reducing power. Furthermore, the effect of chitosan dimers to scavenge different free radicals is different. This may be related to their mechanism. These results indicate that short-chain COS has a strong antioxidant activity and short-chain COS has the potential ability to deal with oxidative stress-related diseases and is expected to develop into a natural antioxidant.

## Figures and Tables

**Figure 1 marinedrugs-19-00366-f001:**
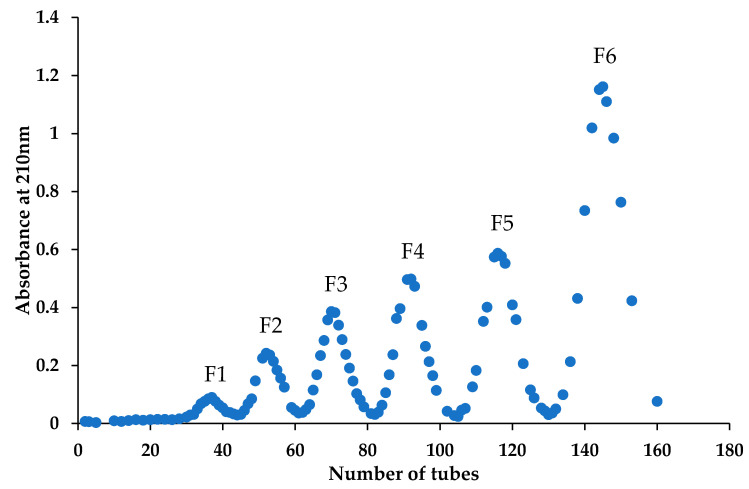
Separation of the chitin oligomers by SEC.

**Figure 2 marinedrugs-19-00366-f002:**
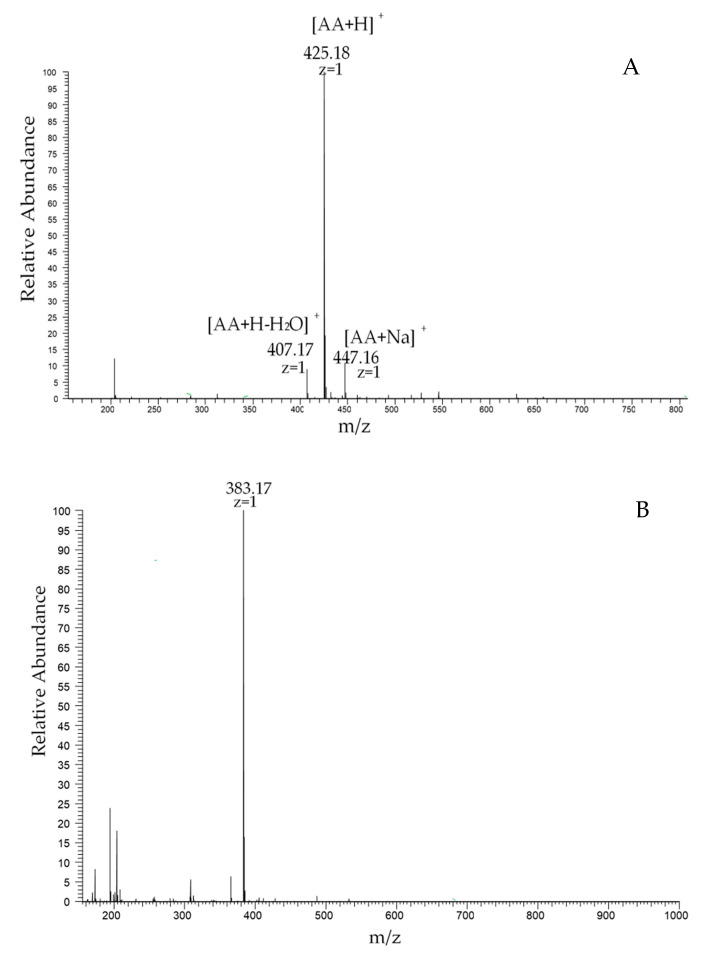
(**A**) The positive-ion mode ESI-MS spectrum of **AA** (F5). The main peak of 425.18 corresponds to the [M + H] + ion-peaks of **AA**. The peak of 407.17 corresponds to the [M + H] + ion-peaks of **AA** by loss of H_2_O. The peak of 447.16 corresponds to the [M + Na] + ion-peaks of **AA**. (**B**) The positive-ion mode ESI-MS spectrum of the N-chitosan dimer. (**C**) is the positive-ion mode ESI-MS spectrum of the V-chitosan dimer; the peak of 383 corresponds to the [M + H] + ion-peaks of **AA** by loss of 42 Da. The peak of 405.15 corresponds to the [M + Na] + ion-peaks of **AA** by loss of 42 Da.

**Figure 3 marinedrugs-19-00366-f003:**
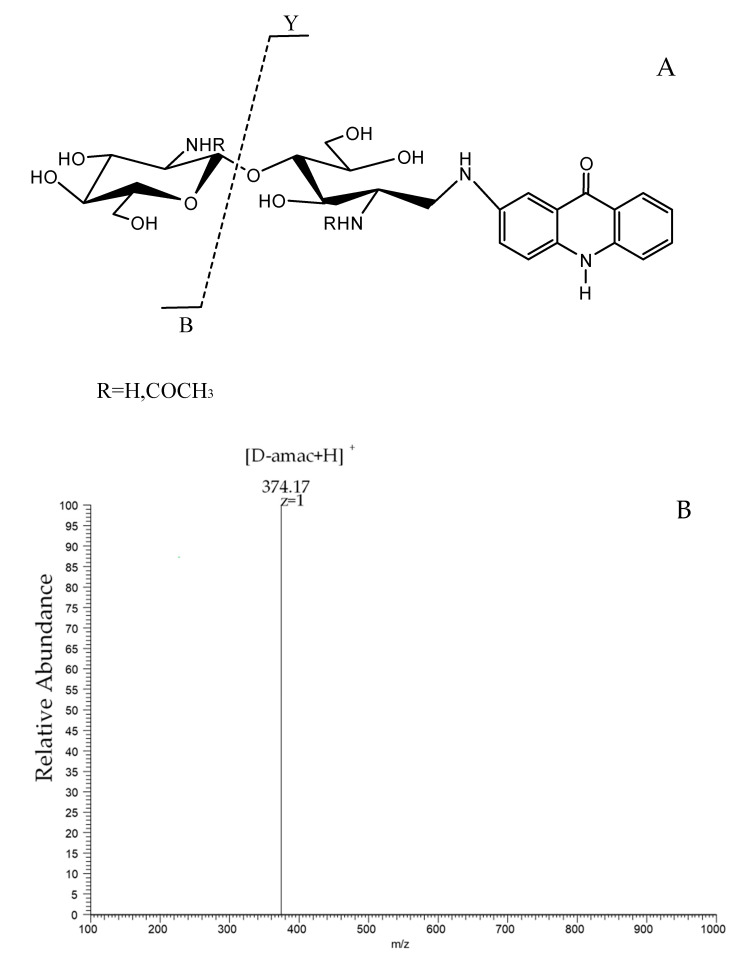
(**A**) The ESI-MS/MS fragmentation of the chitosan dimer. The reductive amination of the chitosan dimer is performed with 2-aminoacridone (amac). Fragmentation of the oligosaccharides leads to the B-type ions from the non-reducing end, and to the Y-type ions from the reducing end. (**B**) The positive-ion mode ESI-MS/MS spectrum of the derivatized V-chitosan dimer at *m*/*z* 577.16. The peak of 374.17 corresponds to the *m*/*z* value of the **D**-amac fragment (the Y-type ions from the reducing end of **AD**). (**C**) The positive-ion mode ESI-MS/MS spectrum of the derivatized N-chitosan dimer at *m*/*z* 577.16. The peak at 416.18 corresponds to the *m*/*z* value of the **A**-amac fragment (Y-type ions from the reducing end of **DA**).

**Figure 4 marinedrugs-19-00366-f004:**
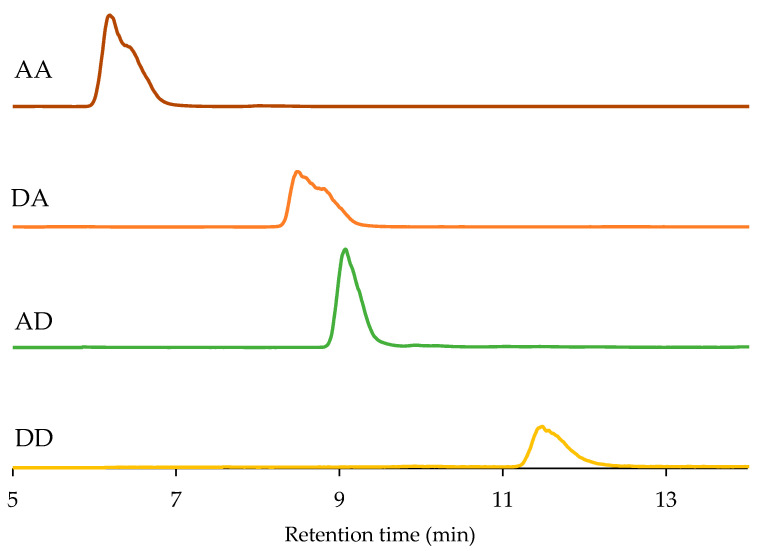
The HPLC spectra of all four chitosan dimers.

**Figure 5 marinedrugs-19-00366-f005:**
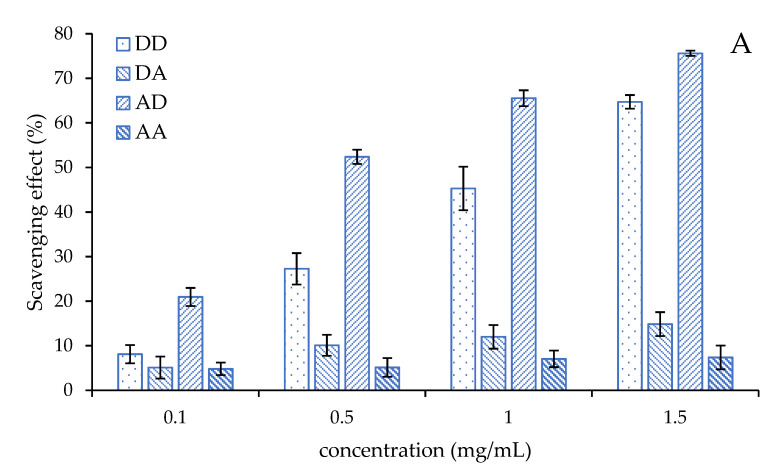
The antioxidant activities of the four chitosan dimers. (**A**) The scavenging effect of the superoxide radical; (**B**) the reducing power; (**C**) the scavenging effect of the hydroxyl radical; (**D**) the scavenging effect of DPPH.

**Table 1 marinedrugs-19-00366-t001:** Characterization of four sequences of the chitosan dimers.

Sequence	Retention Time (min)	Purity	The [M + H] ^+^ Ion
**AA**	6.189	98.9%	425.18
**DA**	8.489	98.0%	383.17
**AD**	9.074	98.1%	383.17
**DD**	11.483	98.4%	341.16

## Data Availability

Not applicable.

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
