# Peer review of "Preparation and Antioxidant Activity of Chitosan Dimers with Different Sequences"

_marinedrugs, 2021, doi:10.3390/md19070366_

Round 1

Reviewer 1 Report

Dear editor:

I consider that the Manuscript number: marinedrugs-1272649, entitled “Enzymatic preparation and antioxidant activity of chitosan dimers with different sequences” is generally interesting and well performed. I think it is suitable for publication in this journal.

Best regards

Author Response

Thank you very much for taking your time to read our article and give it affirmation.

Reviewer 2 Report

In this MS, authors investigate the in-vitro antioxidant properties of different chitosan dimers and discuss their structure-activity relationship. This study is interesting but before it can be considered for publication, there are a few points that need to be clarified in more detail.
- page 3 - the statement that the fraction F5 is a dimer needs to be a little better justified. It is probably true but the authors should bring further evidence beyond what is present in the manuscript.
-page 8 - in Fig. 4 the HPLC spectra of DA and AD dimers partially overlap. what is the meaning of this fact?
- page 10 - fig. 5 shows the antioxidant activities of the four different chitosan dimers. These results, which summarize the meaning of the whole manuscript, should be emphasized.
page 10 - section 3 - the entire first part of this section is a mere repetition of what has already been said in the introduction so that this part should be eliminated. The second part does not illustrate sufficiently well the results obtained but limits itself to formulate hypotheses a little too vague (even if sometimes supported by previous works appeared in the literature). 
Section 4 deals with the experimental part and describes the methods that were used. I don't have criticisms to make or suggestions to give. Everything is described quite clearly. 
In conclusion, it seems to me that the experimental results obtained (even if of limited importance) deserve more than the authors seem to believe or what they discuss in the conclusions.

Author Response

- page 3 - the statement that the fraction F5 is a dimer needs to be a little better justified. It is probably true but the authors should bring further evidence beyond what is present in the manuscript.

Reply: Thanks for giving us the comments. We have been added more detail as is shown in line 94-100 in red. In this study, we separated the N-acetyl chitooligomers mixture [(GlcNAc)1–6] which has six certain components using SEC and we correspondingly obtained six fractions.  Therefore, according to the principle of SEC we speculated F5 is AA firstly and also verified our guess by Electron Spray Ionization/Mass spectrometry (ESI-MS) and High-Performance Liquid Chromatography (HPLC).

-page 8 - in Fig. 4 the HPLC spectra of DA and AD dimers partially overlap. what is the meaning of this fact?

Reply: Thanks for your question. DA and AD are isomers. They both contain a D-glucosamine (GlcN, D) and a β-1,4-linked N acetyl-D-glucosamine (GlcNAc, A), which means their molecule weight and charge are same. At present, it is still very hard to separate oligosaccharides isomers (as mentioned in line72-73). This is why we use the enzymatic method to prepare these two chitosan dimers instead of separation method. We use HPLC mainly to detect their purity, not qualitative analysis. The qualitative analysis is mainly based on a combination of primary mass spectrometry and secondary mass spectrometry (as shown in Figure 2 and 3). The inconsistency of their retention time may be caused by the slightly different affinity between the DA or AD and the column caused by the position of the amino group.

-- page 10 - fig. 5 shows the antioxidant activities of the four different chitosan dimers. These results, which summarize the meaning of the whole manuscript, should be emphasized.

Reply: Thanks for giving us the comments. We have refined this part under your suggestion in line175-176, 183-184, 202 in red. At the same time, we mainly discussed these results in the follow-up discussion part.

page 10 - section 3 - the entire first part of this section is a mere repetition of what has already been said in the introduction so that this part should be eliminated. The second part does not illustrate sufficiently well the results obtained but limits itself to formulate hypotheses a little too vague (even if sometimes supported by previous works appeared in the literature).

Reply: Thanks for giving us the comments. We have eliminated the first part of this section as you suggested. At the same time, we reorganized the second part and added some details in this part in red and removed some ambiguous guesses. Thanks again for your suggestions.

Reviewer 3 Report

The manuscript reports on the observation of the antioxidant properties of four chitosan dimers. The authors have developed the manuscript into an exceptionally well-structured and clear work. A very decent discussion has also given by the author. The reports may give a valuable contribution to the field's knowledge development. However, there are several suggestions for the authors to improve the quality of the manuscript.

Title
The focus of the study is not about the chosen preparation method (enzymatic), therefore putting it in the title is not relevant. Moreover, it is also only one of the techniques used for dimer preparation. 

Abstract
- 16: The information is confusing. The author mentioned "three chitosan dimers" but later "all four chitosan dimers" was mentioned. Please correct this sentence.

Introduction
- 79: Please refer to what AA, AD, and DD stand for as it has not previously explained.
- 81: The author misses the 'in' word before vitro.

General
- A clear elaboration on the differences between each dimer needs to be clearly explained.

Author Response

- Title

The focus of the study is not about the chosen preparation method (enzymatic), therefore putting it in the title is not relevant. Moreover, it is also only one of the techniques used for dimer preparation.

Reply: Thanks for giving us the comments. We seriously considered your suggestion and changed the title to “Preparation and Antioxidant Activity of Chitosan Dimers with Different Sequences”.

- Abstract

- 16: The information is confusing. The author mentioned "three chitosan dimers" but later "all four chitosan dimers" was mentioned. Please correct this sentence.

Reply: Thanks for giving us the comments. In this study, we only prepared three chitosan dimers (AA, AD, DA) and DD was prepared by ion-exchange chromatography as reported previously in our laboratory (as mentioned in line 81 and 150). Then we studied the antioxidant activity of all these sequences.

- 79: Please refer to what AA, AD, and DD stand for as it has not previously explained.

Reply: Thanks for giving us the comments. We think your proposal is very appropriate, so we added a footnote on line 46 when we first introduced the glucose unit. In this way the meanings of AA, AD, DA and DD are clear.

- 81: The author misses the 'in' word before vitro.

Reply: Thank you for pointing out this mistake, we have revised this and thank you again for your careful review.

Round 2

Reviewer 2 Report

The new version of the manuscript takes into account the suggestions I made in my previous report. The authors have responded satisfactorily to my comments and I believe that the manuscript can now be published.